# ElliCE: Efficient and Provably Robust Algorithmic Recourse via the Rashomon Sets

**Bohdan Turbal**[1]    **Iryna Voitsitska**[2]    **Lesia Semenova**[3*]

[1] Princeton University [2] Ukrainian Catholic University [3] Rutgers University
bt4811@princeton.edu, voitsitska.pn@ucu.edu.ua, lesia.semenova@rutgers.edu

## Abstract

Machine learning models now influence decisions that directly affect people's lives, making it important to understand not only their predictions, but also how individuals could act to obtain better results. Algorithmic recourse provides actionable input modifications to achieve more favorable outcomes, typically relying on counterfactual explanations to suggest such changes. However, when the Rashomon set – the set of near-optimal models – is large, standard counterfactual explanations can become unreliable, as a recourse action valid for one model may fail under another. We introduce ElliCE, a novel framework for robust algorithmic recourse that optimizes counterfactuals over an ellipsoidal approximation of the Rashomon set. The resulting explanations are provably valid over this ellipsoid, with theoretical guarantees on uniqueness, stability, and alignment with key feature directions. Empirically, ElliCE generates counterfactuals that are not only more robust but also more flexible, adapting to user-specified feature constraints while being substantially faster than existing baselines. This provides a principled and practical solution for reliable recourse under model uncertainty, ensuring stable recommendations for users even as models evolve.

## 1 Introduction

When an algorithmic decision denies someone a loan, a job, or insurance coverage, a natural question follows: *What could I change to obtain a better outcome next time?* Algorithmic recourse answers this question by providing concrete, actionable changes that could lead to a more favorable decision. A common way to generate such recommendations is through counterfactual explanations—small modifications to an individual's features that flip the model's prediction. Yet, even when the recommendation looks specific (e.g. "increase your income by $5000"), one must ask: *"Would that same change still work tomorrow if the institution retrains or replaces its model?"* or *"How stable are these suggestions across equally good models that explain the data in different ways?"*

Most existing counterfactual generation methods [43, 46, 48, 50, 52, 55, 59, 63] implicitly assume that the underlying model is fixed and perfectly known. In practice, models evolve: banks regularly retrain risk predictors, healthcare institutions update diagnostic classifiers, and regulators may require model re-validation under new privacy or transparency constraints. Small shifts in data or regularization can result in very different-yet-equally-accurate models. This phenomenon, known as the Rashomon Effect [9, 15, 23, 54, 56], implies that many distinct predictors achieve nearly optimal performance. In such settings, a counterfactual valid for one model can fail under another, undermining the reliability and consistency of algorithmic recourse.

---

*The majority of this work was conducted while Bohdan was at Taras Shevchenko National University of Kyiv and Lesia at Microsoft Research NYC.

Recent approaches have attempted to produce robust counterfactuals, meaning counterfactuals that are valid under small parameter perturbations or across predefined ensembles [19, 22, 26, 32, 33, 34, 38, 61]. However, these methods either rely on heavy-weight mixed-integer solvers, restrict robustness to local neighborhoods around a single model, or lack formal guarantees of validity across the full space of near-optimal solutions known as the Rashomon set. None of them directly leverages the geometry of this Rashomon set itself.

We introduce ElliCE, an efficient and provably robust framework for algorithmic recourse that works over an ellipsoidal approximation of the Rashomon set. By modeling the space of near-optimal models as an ellipsoid derived from the curvature (Hessian) of the loss landscape, ElliCE reformulates robust counterfactual generation as a tractable convex optimization problem. The resulting counterfactuals are valid for every model inside the ellipsoid, ensuring that a user's recommended action remains meaningful even if the deployed model is replaced by another equally accurate one from the approximated Rashomon set.

Our contributions are fourfold: (1) *Theoretical foundation*. We derive a closed-form expression for the worst-case prediction, which allows us to formulate the robust recourse problem as a convex optimization and establish formal guarantees of validity, uniqueness, and stability for ElliCE's counterfactuals. (2) *Geometric intuition*. We show that ElliCE's robustness term connects the counterfactual's stability with the importance of the features it modifies as the optimization naturally aligns recourse directions with the principal curvature axes of the loss landscape. (3) *Actionability*. ElliCE supports feature-level constraints, such as sparsity constraints, immutable or range-restricted attributes, allowing users to generate realistic, actionable recourse tailored to specific application or user settings. (4) *Empirical validation*. Across multiple high-stakes tabular datasets, ElliCE achieves higher robustness and remains one to three orders of magnitude faster than competing baselines, while maintaining proximity and plausibility.

Ultimately, ElliCE looks at algorithmic recourse through the lens of model multiplicity. Instead of relying on a single model's decision boundary, it offers explanations that stay consistent across many models that fit the data almost equally well. This perspective treats the Rashomon Effect not as a flaw to eliminate, but as an inherent source of uncertainty to account for, leading to stable recourse in the presence of model diversity.

## 2   Related works

**Rashomon Effect.** The Rashomon Effect, a term popularized by Breiman [9] in the context of machine learning, describes the phenomenon where multiple distinct models can achieve near-optimal empirical risk (these models form a Rashomon set). This effect is also referred to as model multiplicity [6, 45]. The existence of the Rashomon set has implications for the trustworthiness and reliability of machine learning systems, influencing feature importance [16, 17, 21, 49], fairness [14, 42, 47], the existence of simple yet accurate models [7, 56, 57] to name a few. Significant research has focused on measuring and characterizing the Rashomon set for different model classes [28, 29, 30, 64, 66]. Our work leverages insights into the geometry of the Rashomon set, explored by works like Donnelly et al. [18], Zhong et al. [66], but applies them to the distinct challenge of generating robust algorithmic recourse across this set.

**Counterfactual Explanations (CEs).** Counterfactual Explanations have emerged as a prominent tool for providing algorithmic recourse. Numerous approaches exist for generating CEs. Proximity-based methods aim for counterfactuals requiring minimal feature space perturbations [10, 48, 62, 63]. Sparsity techniques prioritize modifying the fewest features possible to enhance actionability [46, 59], while some methods attempt to balance both objectives [43]. Another research direction emphasizes plausibility, ensuring generated CEs represent realistic instances by constraining them to the data manifold, for example, using guidance from generative models [36, 50, 51], encoding feasibility rules [37], or tracing density-aware paths [52]. Recent extensions also incorporate temporal reasoning [12] and fairness objectives [5, 40, 65]. A key limitation across these approaches (which ElliCE directly addresses) is the assumption of a fixed, perfectly known predictive model, as counterfactuals constructed near a specific decision boundary can become unstable under model updates or perturbations.

**Robustness to Local Model Perturbations.** Building upon the limitation of fixed models, one line of work has focused specifically on achieving robustness against small, local changes or per-

turbations in the model's parameters. For instance, ROAR [61] optimizes CEs considering local $\Delta$-perturbations of the model. Jiang et al. [31] introduced $\Delta$-robustness, a formal metric to assess CE validity under bounded parameter perturbations in neural networks, with subsequent works developing provably robust MILP-based methods [32]. While these methods offer formal guarantees for $\Delta$-robustness, MILP-based approaches can face scalability challenges, and the focus is generally on local parameter stability rather than the broader implications of the Rashomon Effect.

**Robustness under the Rashomon Effect.** A growing body of work addresses counterfactual robustness under model multiplicity, aligning closely with the Rashomon Effect. Several approaches evaluate stability across predefined sets or ensembles of models, introducing heuristic stability measures (e.g., T:Rex [26] and RobX [19]), probabilistic frameworks [22, 38], or guarantees under specific norms and conditions like distribution shift [24, 39, 41]. Foundational work by Pawelczyk et al. [51] conceptually linked the Rashomon Effect to counterfactuals, though primarily enhancing input perturbation robustness. More recent methods use argumentative ensembling [34] or aggregate explanations across AutoML-generated sets [11] to handle model multiplicity.

Our work takes a distinct approach. Rather than relying on ensemble agreement, heuristic stability metrics, local perturbations, or argumentative aggregation, ElliCE leverages the local geometry of the Rashomon set, approximated by an ellipsoid, to derive theoretically grounded, robust recourse valid across all models within the approximation.

## 3 Background and Notation

**Dataset and hypothesis space**. Consider $n$ i.i.d. samples $\mathcal{S}_n = \{\mathbf{z}_i = (\mathbf{x}_i, y_i)\}_{i=1}^n$, where $\mathbf{x}_i \in \mathcal{X} \subseteq \mathbb{R}^d$ and $y_i \in \mathcal{Y} = \{0, 1\}$ are generated from an unknown distribution $\mathcal{D}$ on $\mathcal{X} \times \mathcal{Y}$. Let $\mathcal{Y}_{pred}$ be an output space, where $\mathcal{Y}_{pred} \subseteq \mathbb{R}$ for scores (logits) or $\mathcal{Y}_{pred} \subseteq [0, 1]$ for probabilities. Then $\mathcal{F} = \{f_{\boldsymbol{\theta}} : \boldsymbol{\theta} \in \Theta\}$ is a hypothesis space of functions $f_{\boldsymbol{\theta}} : \mathcal{X} \to \mathcal{Y}_{pred}$, parameterized by a vector $\boldsymbol{\theta} \in \Theta \subseteq \mathbb{R}^p$. For example, $\mathcal{F}$ can represent linear models or multilayer perceptrons. We denote a specific function by $f_{\boldsymbol{\theta}}$. As our analysis focuses on the parameter space $\Theta$, we will often refer to the model directly by its parameter vector $\boldsymbol{\theta}$.

**Loss and objective function**. Let $\phi : \mathcal{Y}_{pred} \times \mathcal{Y} \to \mathbb{R}_+$ be a loss function. In this work, we consider binary cross-entropy (logistic) loss $\phi(f_{\boldsymbol{\theta}}(\mathbf{x}), y) = -[y \log(\sigma(f_{\boldsymbol{\theta}}(\mathbf{x}))) + (1 - y) \log(1 - \sigma(f_{\boldsymbol{\theta}}(\mathbf{x})))]$, which is applied to the model's raw score (logit), $s = f_{\boldsymbol{\theta}}(\mathbf{x})$, where $\sigma(s) = \frac{1}{1+\exp(-s)}$ is the sigmoid function. However, our results generalize to other convex losses. The true risk is the expected loss $J(\boldsymbol{\theta}) = \mathbb{E}_{\mathbf{z}}[\phi(f_{\boldsymbol{\theta}}(\mathbf{x}), y)]$ that we approximate with the empirical risk, which is the average loss, $\hat{J}(\boldsymbol{\theta}) = \frac{1}{n} \sum_{i=1}^n \phi(f_{\boldsymbol{\theta}}(\mathbf{x}_i), y_i)$. We also define an $\ell_2$-regularized objective function: $\hat{L}(\boldsymbol{\theta}) = \hat{J}(\boldsymbol{\theta}) + \frac{\lambda}{2}||\boldsymbol{\theta}||_2^2$, where $\lambda \geq 0$ is the regularization strength. The empirical risk minimizer (ERM) is $\hat{\boldsymbol{\theta}} \in \arg\min_{\boldsymbol{\theta} \in \Theta} \hat{L}(\boldsymbol{\theta})$. When $\lambda = 0$, the ERM is $\hat{\boldsymbol{\theta}} \in \arg\min_{\boldsymbol{\theta} \in \Theta} \hat{J}(\boldsymbol{\theta})$.

**Rashomon set**. Following [21, 56, 64], we define the $\epsilon$-Rashomon set within the parameter space $\Theta$ as the set of parameter vectors whose corresponding models $f_{\boldsymbol{\theta}}$ have objective value close to the minimum:
$$\mathcal{R}(\epsilon) := \{\boldsymbol{\theta} \in \Theta : \hat{L}(\boldsymbol{\theta}) \leq \hat{L}(\hat{\boldsymbol{\theta}}) + \epsilon\},$$
where $\epsilon \geq 0$ is the Rashomon parameter defining the allowable tolerance in objective compared to the ERM, $\hat{L}(\hat{\boldsymbol{\theta}})$. It is typically a small value. The existence of the Rashomon set with multiple, distinct parameter vectors $\boldsymbol{\theta}$ (corresponding to potentially distinct functions $f_{\boldsymbol{\theta}}$) achieving similar performance implies that different underlying logic (how features contribute to predictions) can explain the data equally well. It is important to be aware of this variability among near-optimal models when generating explanations for individual predictions, as different models in $\mathcal{R}(\epsilon)$ might suggest different ways an outcome could be changed.

**Counterfactual explanations.** Let $g : \mathcal{Y}_{pred} \to \{0, 1\}$ be the decision function that converts a model's score output $s = f_{\boldsymbol{\theta}}(\mathbf{x})$ to a final binary class label by applying a threshold $t$, such that $g(s) = \mathbf{1}[s \geq t]$. For an ERM $\hat{\boldsymbol{\theta}}$ and for an input vector $\mathbf{x}_0$ with prediction $g(f_{\hat{\boldsymbol{\theta}}}(\mathbf{x}_0)) = \hat{y}_0$, a counterfactual explanation $\mathbf{x}_c$ is a data point such that its predicted class is the opposite, i.e., $g(f_{\hat{\boldsymbol{\theta}}}(\mathbf{x}_c)) = 1 - \hat{y}_0$. The set of all counterfactual explanations for $\mathbf{x}_0$ under the model $\hat{\boldsymbol{\theta}}$ and decision function $g$ is defined as:
$$\mathcal{C}(\mathbf{x}_0, \hat{\boldsymbol{\theta}}) = \{\mathbf{x}_c \in \mathcal{X} : g(f_{\hat{\boldsymbol{\theta}}}(\mathbf{x}_c)) = 1 - g(f_{\hat{\boldsymbol{\theta}}}(\mathbf{x}_0))\}.$$

For instance, in a credit loan application scenario, if an applicant $\mathbf{x}_0$ is denied a loan (e.g., $g(f_{\hat{\boldsymbol{\theta}}}(\mathbf{x}_0)) = 0$), a counterfactual explanation $\mathbf{x}_c$ would be a modified version of their application details (e.g., increased income, reduced debt) such that the model predicts approval, $g(f_{\hat{\boldsymbol{\theta}}}(\mathbf{x}_c)) = 1$. While many such $\mathbf{x}_c$ might exist, practical algorithmic recourse aims to find explanations that require minimal change for the user. This means finding the "closest" counterfactual: $\mathbf{x}_c^* = \arg\min_{\mathbf{x}_c \in \mathcal{C}(\mathbf{x}_0, \hat{\boldsymbol{\theta}})} \nu(\mathbf{x}_c, \mathbf{x}_0)$, where $\nu(\cdot, \cdot) : \mathbb{R}^d \times \mathbb{R}^d \to \mathbb{R}$ is a defined distance function or cost metric that we discuss next.

**Distance Metrics.** In our framework, we primarily focus on the two distance metrics for generating actionable and interpretable counterfactuals: $\ell_2$ or Euclidean distance and mixed distance $\ell_{mix}$. Note that $\ell_2$ is a natural geometric measure of proximity, that penalizes large differences in any feature, $\nu(\mathbf{x}_c, \mathbf{x}_0) = \ell_2(\mathbf{x}_c, \mathbf{x}_0) = \|\mathbf{x}_c - \mathbf{x}_0\|_2^2 = \sum_{j=1}^d (x_{cj} - x_{0j})^2$. For practical applications where features have different natures (continuous and categorical), one can also consider the mixed distance $\nu(\cdot, \cdot) = \ell_{mix}$, inspired by Gower's distance. Assuming that the data are standardized, it is defined as: $\ell_{mix}(\mathbf{x}_c, \mathbf{x}_0) = \sqrt{\sum_{j \in \mathcal{I}_{cont}} (x_{cj} - x_{0j})^2 + \sum_{j \in \mathcal{I}_{cat}} \bar{u}_j \mathbf{1}[x_{cj} \neq x_{0j}]}$, where $\mathcal{I}_{cont}$ and $\mathcal{I}_{cat}$ denote the sets of continuous and categorical feature indices respectively, $\mathbf{1}[\cdot]$ is the indicator function, and $\bar{u}_j$ are optional weights reflecting the cost of changing feature $j$. We use $\ell_2$ distance for our theoretical analysis in the subsequent sections.

Next, we describe our approximating framework and outline the optimization process.

## 4 A Framework for Robust Recourse over the Rashomon Set

We focus our theoretical analysis on linear predictors of the form $f_{\boldsymbol{\theta}}(\mathbf{x}) = \boldsymbol{\theta}^\top \mathbf{x}$. However, the same methodology applies in the final embedding space of multilayer perceptrons (MLPs) by writing the model as $f_{\boldsymbol{\theta}}(\mathbf{x}) = \boldsymbol{\theta}^\top h(\mathbf{x})$, where $h(\mathbf{x})$ is the penultimate-layer embedding and $\boldsymbol{\theta}$ are the last-layer parameters. We freeze $h(\cdot)$ and apply the same ellipsoidal procedure to $\boldsymbol{\theta}$ as in the linear case (equivalently, replace $\mathbf{x}$ by $h(\mathbf{x})$ in the formulas below).

**Approximated Rashomon set.** For certain objectives, such as $\ell_2$-regularized mean-squared error on linear models, the Rashomon set is exactly an ellipsoid in the parameter space [56]: $\mathcal{R}(\epsilon) = \{\boldsymbol{\theta} : (\boldsymbol{\theta} - \hat{\boldsymbol{\theta}})^\top (X^\top X + \lambda I_p)(\boldsymbol{\theta} - \hat{\boldsymbol{\theta}}) \leq \epsilon\}$, where $X \in \mathbb{R}^{n \times d}$ is the data matrix, whose $i$-th row is the feature vector $\mathbf{x}_i^\top$, $I_p$ is an identity matrix of size $p \times p$, and $\lambda \in \mathbb{R}_+$ is the regularization parameter. Because mean-squared error provides a local quadratic approximation to other convex losses, the exact ellipsoidal form of its Rashomon set serves as strong motivation for the Rashomon set approximation. Building on this and on similar geometric intuition [66], we approximate the $\epsilon$-Rashomon set with an ellipsoid defined by the local geometry of the loss landscape:

$$\hat{\mathcal{R}}(\epsilon) = \{\boldsymbol{\theta} : \frac{1}{2}(\boldsymbol{\theta} - \hat{\boldsymbol{\theta}})^\top H (\boldsymbol{\theta} - \hat{\boldsymbol{\theta}}) \leq \epsilon\},$$

where $H = X^\top W X + \lambda I_p$ is the Hessian of the $\ell_2$-regularized loss function, evaluated at $\hat{\boldsymbol{\theta}}$. For logistic loss, $W$ is an $n \times n$ diagonal matrix of weights where $w_{ii} = \sigma(f_{\hat{\boldsymbol{\theta}}}(\mathbf{x}_i))(1 - \sigma(f_{\hat{\boldsymbol{\theta}}}(\mathbf{x}_i)))$. Recall from Section 3 that $\sigma(\cdot)$ is the sigmoid function.

The Hessian matrix $H$ of the regularized objective $\hat{L}(\boldsymbol{\theta})$ is strictly positive definite. This is because it is the sum of the positive semidefinite (PSD) Hessian from the convex logistic loss and the positive definite (PD) Hessian from the $\ell_2$ regularization term ($\lambda I_p$), assuming $\lambda > 0$. A positive definite Hessian is important for our framework, as it guarantees the approximating ellipsoid is well-defined and bounded, and ensures that $H$ is invertible for our closed-form solution.

In cases where the unregularized risk $\hat{J}(\boldsymbol{\theta})$ is minimized (e.g., for neural networks), the resulting Hessian is only guaranteed to be PSD and may be singular. For these models, we ensure positive definiteness in practice by adding a small stabilization term, $\alpha I_p$, $\alpha > 0$, to the computed Hessian, which is a standard technique to guarantee invertibility.

**Optimization.** To find a robust counterfactual explanation, we want to compute an explanation $\mathbf{x}_c$ that is closest to the original point $\mathbf{x}_0$ while ensuring that its predicted outcome is above a target threshold $t$ for all models within the approximated Rashomon set. In other words, for a given $\mathbf{x}_0$, we look for a minimally modified (measured in some distance; we will use $\ell_2$ here) $\mathbf{x}_c$, such that its predicted outcome achieves $t$ even when evaluated by the least favorable model $\boldsymbol{\theta}$ within the

approximated Rashomon set $\hat{\mathcal{R}}(\epsilon)$. Formally, this requirement leads to the following optimization problem:

$$\min_{\mathbf{x}_c} \quad \|\mathbf{x}_c - \mathbf{x}_0\|_2^2 \qquad \text{s.t.} \quad \min_{\boldsymbol{\theta} \in \hat{\mathcal{R}}(\epsilon)} \boldsymbol{\theta}^\top \mathbf{x}_c \geq t. \tag{1}$$

The inner minimization problem admits a closed-form solution, as we show next in Theorem 1. By reformulating the problem in this way, we get a tractable optimization framework that supports more efficient computation and analytical analysis of solution properties.

**Theorem 1** (Closed-form solution)**.** *For positive-definite Hessian $H$, the inner minimization problem over the ellipsoid-approximated Rashomon set $\hat{\mathcal{R}}(\epsilon)$ has the closed-form solution $\min_{\boldsymbol{\theta} \in \hat{\mathcal{R}}(\epsilon)} \boldsymbol{\theta}^\top \mathbf{x}_c = \hat{\boldsymbol{\theta}}^\top \mathbf{x}_c - \sqrt{2\epsilon\, \mathbf{x}_c^\top H^{-1} \mathbf{x}_c}$. Moreover, for a given $\mathbf{x}_c$, the worst-case model $\boldsymbol{\theta}_{worst}(\mathbf{x}_c)$ that achieves this minimum is: $\boldsymbol{\theta}_{worst}(\mathbf{x}_c) = \hat{\boldsymbol{\theta}} - \sqrt{2\epsilon}\, \frac{H^{-1}\mathbf{x}_c}{\sqrt{\mathbf{x}_c^\top H^{-1} \mathbf{x}_c}}$.*

We prove Theorem 1 in Appendix A.1. As a direct consequence of Theorem 1, we obtain a practical criterion for verifying the robustness of a potential counterfactual. Specifically, since the theorem provides an explicit characterization of the output generated by the least favorable model $\boldsymbol{\theta} \in \hat{\mathcal{R}}(\epsilon)$ for a given $\mathbf{x}_c$, we can immediately determine if this $\mathbf{x}_c$ achieves the target $t$ across the entire set as we show in the following corollary.

**Corollary 1.** *A given counterfactual explanation $\mathbf{x}_c$ is robust with respect to all models in the ellipsoid-approximated Rashomon set $\hat{\mathcal{R}}(\epsilon)$ against a target score $t$ if and only if: $\hat{\boldsymbol{\theta}}^\top \mathbf{x}_c - \sqrt{2\epsilon\, \mathbf{x}_c^\top H^{-1} \mathbf{x}_c} \geq t$.*

By substituting the closed-form solution from Theorem 1 into the original optimization problem (1), the robust counterfactual optimization problem becomes:

$$\min_{\mathbf{x}_c} \quad \|\mathbf{x}_c - \mathbf{x}_0\|_2^2 \qquad \text{s.t.} \quad \hat{\boldsymbol{\theta}}^\top \mathbf{x}_c - \sqrt{2\epsilon\, \mathbf{x}_c^\top H^{-1} \mathbf{x}_c} \geq t. \tag{2}$$

The resulting problem is a quadratically constrained quadratic program (QCQP), which is a class of tractable convex optimization problems. We solve it efficiently using a gradient-based method. Leveraging the formulation (2), we implement two approaches for generating counterfactuals: a search-based method for generating data-supported counterfactuals lying on the data manifold, and a continuous optimization method for exploring potentially novel non-data supported solutions.

**Continuous CE generation.** For non-data supported counterfactuals, we solve the convex optimization problem in Equation (2) using a gradient-based approach for both linear models and multilayer perceptrons. This method directly optimizes for a counterfactual $\mathbf{x}_c$ in the input space. For neural networks, the process is guided by the worst-case model $\boldsymbol{\theta}_{worst}(\mathbf{x}_c)$ identified in the final layer's embedding space using Theorem 1, with the resulting gradients mapped back to the input features. The full details of this procedure are available in Appendix B.4.

**Data-supported CE generation**. For practical applications where counterfactuals should remain on the data manifold, we generate data-supported explanations based on the training set. Specifically, we evaluate the robust logit $\hat{\boldsymbol{\theta}}^\top \mathbf{x}_i - \sqrt{2\epsilon \mathbf{x}_i^\top H^{-1} \mathbf{x}_i}$ for each training data point $\mathbf{x}_i$ using Theorem 1. Then, we compute the subset $S_{stable}$ by filtering out points where this robust prediction exceeds the target threshold $t$. Finally, we use k-d tree nearest neighbor search within $S_{stable}$ to identify the points closest to the input point $\mathbf{x}_0$ in terms of defined distance (for example, $\ell_2$), which gives us a counterfactual that is both robust and lies on the data manifold.

The continuous approach offers flexibility by exploring the entire feature space for new solutions, while the data-supported approach guarantees plausibility by restricting solutions to observed examples. We evaluate the performance of both approaches in Section 6 and focus on theoretical guarantees of our framework next.

## 5 Theoretical Guarantees of ElliCE Counterfactuals

In this section, we explore key theoretical properties of the counterfactual explanations generated under our framework. Note that we use $\ell_2$ distance as target distance between $\mathbf{x}_0$ and $\mathbf{x}_c$. We show that the counterfactual explanations generated by our method are valid, unique, stable, and align

with important directions in the feature space. We focus on each of these properties separately and proofs of theorems provided in this section are in Appendix A.2.

**Validity.** By explicitly optimizing for the worst-case model $\boldsymbol{\theta}_{worst}$ within the defined ellipsoid, any counterfactual $\mathbf{x}_c$ generated by ElliCE is, by construction, valid for all models in the approximated Rashomon set. This inherent validity ensures that the provided recourse is faithful, regardless of which model from the approximated Rashomon set was selected.

**Uniqueness.** By Theorem 2, that we state next, any solution $\mathbf{x}_c$ to the optimization problem (2) is unique. Because our objective is strictly convex and the approximated Rashomon set is characterized as an ellipsoid, for a given $\mathbf{x}_0$, there can never be two distinct counterfactuals at the same $\ell_2$ distance from the original $\mathbf{x}_0$. In practical terms, this uniqueness guarantees that ElliCE provides a single solution for a given input and desired robustness level. This directly addresses and resolves "explanation multiplicity" [25], where multiple, distinct explanation paths might exist for a single input (at least for $\ell_2$ distance).

**Theorem 2** (Uniqueness). *If a solution $\mathbf{x}_c$ to the optimization problem (2) exists, then $\mathbf{x}_c$ is unique.*

**Stability.** The input data $\mathbf{x}_0$ is often subject to noise or minor variations. A desirable property is that such small changes in the input do not lead to drastically different counterfactuals. Our framework ensures this stability. Theorem 3 formally states that the process of generating robust counterfactuals is Lipschitz continuous with a constant of 1. This means that if the original input $\mathbf{x}_0$ is perturbed by a small amount $\boldsymbol{\delta}$ to become $\mathbf{x}_0'$, the resulting robust counterfactual $\mathbf{x}_c'$ will not deviate from the original counterfactual $\mathbf{x}_c$ by more than the magnitude of the initial perturbation $\|\boldsymbol{\delta}\|_2$. This property guarantees the reliability of the explanations.

**Theorem 3** (Stability). *Given an input $\mathbf{x}_0$, let $\mathbf{x}_c$ be the robust counterfactual solution for $\mathbf{x}_0$. If the input is perturbed to $\mathbf{x}_0' = \mathbf{x}_0 + \boldsymbol{\delta}$, where $\boldsymbol{\delta} \in \mathbb{R}^d$, and $\mathbf{x}_c'$ is the robust counterfactual solution for $\mathbf{x}_0'$, then $\|\mathbf{x}_c - \mathbf{x}_c'\|_2 \leq \|\boldsymbol{\delta}\|_2$.*

**Alignment with Important Feature Directions.** An insightful explanation should not only provide a path to a different outcome but also highlight which features are most critical in achieving that change, particularly under model uncertainty. The robustness penalty term, $C_{rob}(\epsilon, \mathbf{x}_c) = \sqrt{2\epsilon \mathbf{x}_c^\top H^{-1} \mathbf{x}_c}$, plays a key role in this alignment. Theorem 4 formalizes the intuition that to minimize this penalty (and thus find an efficient robust counterfactual), the recourse direction $\mathbf{x}_c$ should align with directions in the feature space that are most sensitive or influential, as captured by the eigenvectors of the Hessian matrix $H$. Specifically, under reasonable conditions, the penalty is minimized when the counterfactual aligns with the leading eigenvector of $H$, which often corresponds to the direction of greatest sensitivity. This encourages the counterfactual to suggest changes along features that have a significant impact, making the explanation more informative.

**Theorem 4** (Alignment with Important Feature Directions). *Define the robustness penalty as $C_{rob}(\epsilon, \mathbf{x}_c) = \sqrt{2\epsilon \mathbf{x}_c^\top H^{-1} \mathbf{x}_c}$ for a symmetric positive definite Hessian $H$. Let $\lambda_1$ be the largest eigenvalue of $H$ with corresponding eigenvector $\mathbf{q}_1$, and assume that $\lambda_1$ is unique. Then, for a fixed non-zero norm $\|\mathbf{x}_c\|_2$, the robustness penalty term $C_{rob}(\epsilon, \mathbf{x}_c)$ is minimized when the counterfactual vector $\mathbf{x}_c$ is aligned (i.e., collinear) with the eigenvector $\mathbf{q}_1$.*

**Price of robustness.** Previous literature has observed the trade-off between robustness and proximity [22]. Indeed, intuitively, increasing robustness and ensuring validity across a larger set of potential models may require more changes to the input features, effectively increasing the proximity. This implies a "cost" for greater robustness that Theorem 5 formalizes.

**Theorem 5** (Robustness-Proximity Trade-off). *For an input $\mathbf{x}_0$ such that $\hat{\boldsymbol{\theta}}^\top \mathbf{x}_0 \leq t$, where $\hat{\boldsymbol{\theta}}$ is ERM, let $\mathbf{x}_c^*(\epsilon)$ be the optimal robust counterfactual for a given robustness level $\epsilon > 0$, and let $\nu(\epsilon) = \|\mathbf{x}_c^*(\epsilon) - \mathbf{x}_0\|_2^2$ be its $\ell_2$ distance from $\mathbf{x}_0$. If $\nu(\epsilon_1) > 0$ and $\mathbf{x}_c^*(\epsilon_1) \neq \mathbf{0}$, then for any two robustness levels $0 < \epsilon_1 < \epsilon_2$, $\nu(\epsilon_1) < \nu(\epsilon_2)$.*

The practical impact of this trade-off is significant. Overly robust counterfactuals may become distant and unactionable, while insufficient robustness compromises recourse reliability under model shifts. This underscores the need for methods that efficiently explore this trade-off by achieving substantial robustness with reasonable proximity—a goal that ElliCE effectively meets (Figure 2).

When applying our theoretical results to MLPs, *the validity guarantee is fully preserved in the input space*, which is a key result. The formal guarantees for uniqueness (Theorem 2), stability

(Theorem 3), and the robustness-proximity trade-off (Theorem 5), however, depend on the convexity of the feasible set (see proof of Theorem 2). While this convexity is guaranteed in the embedding space $h(\mathbf{x})$, the nonlinear mapping from the input space ($\mathbf{x} \mapsto h(\mathbf{x})$) means it is not guaranteed to hold there. This distinction highlights a fundamental challenge for robust recourse in deep models and underscores that extending these formal guarantees to the input space is a promising direction for future work. Nonetheless, these theorems provide a principled geometric foundation for our approach and hold for linear models and embedding spaces. Next, we present empirical results showing that ElliCE's performance is consistent with its theoretical guarantees.

## 6 Evaluation Pipeline and Experimental Results

In our evaluation pipeline, we work with the hypothesis space of linear models and multi-layer perceptrons (MLPs). However, our results can be extended to other hypothesis spaces that can be optimized with gradient descent, such as neural additive models [1]. In this section, we empirically show that ElliCE is faster and more robust as compared to other methods that produce robust counterfactuals. Please see Appendix B for additional details and results.

**Datasets.** We consider nine datasets from high-stakes decision domains such as lending (Australian Credit [53], FICO [20], German Credit [27], Banknote [44]), healthcare (Parkinson's [60], Diabetes [58]), and recidivism (COMPAS [2]), as well as benchmark datasets (Wine Quality [13], Extended Iris [3]). Please see Table 3 for detailed dataset descriptions and preprocessing notes. We used datasets with predominantly categorical features (FICO, Australian Credit, COMPAS, German Credit, Diabetes) for data-supported CE generation, and datasets with continuous features (Diabetes, Parkinson's, Banknote, Iris, and Wine Quality) for continuous methods. We balanced the datasets, standardized continuous features, and, for some datasets, dropped rows with missing values.

**Baselines.** We compare ElliCE to other methods that are designed to generate robust counterfactual explanations, such as T:Rex, Interval Abstractions (we refer to it as Delta-robustness [31]), PRO-PLACE, and ROAR. *T:Rex* [26] generates robust counterfactuals for neural networks using a Stability measure that depends on variance. It quantifies robustness to naturally occurring model changes, providing probabilistic validity guarantees. It is a successor of RobX [19], which targets tree-based ensembles. *Interval Abstractions* [33] ensures that counterfactuals are robust to bounded changes in model parameters (weights and biases). It uses interval neural networks and mixed-integer linear programming. *PROPLACE* [32] formulates counterfactual generation as a bi-level robust optimization problem: it enforces plausibility by restricting solutions to the convex hull of realistic samples and uses interval bounds on neural networks to ensure robustness. *ROAR* [61] optimizes counterfactual validity under bounded model parameter perturbations using a robustness-constrained loss formulation. Most implementations of our baselines follow Jiang et al. [35].

**Evaluators.** Precisely computing the entire Rashomon set for the hypothesis spaces that we consider is intractable. Therefore, to evaluate the robustness and validity of counterfactual explanations generated by ElliCE and the baselines, we rely on established techniques that approximate or characterize this set. These approaches generate diverse collections of near-optimal models, each serving as a proxy for the actual Rashomon set. Our evaluators include: *Random Retrain*, which retrains models multiple times with different random seeds to capture procedural variability. *Rashomon Dropout* [30], which applies binary dropout masks to a single trained neural network's weights during inference, creating an ensemble of thinned sub-models. *Adversarial Weight Perturbation (AWP)* [28], which generates diverse models from an initially trained model by applying small perturbations to its weights. We define the objective tolerance (Rashomon parameter) for the evaluators as $\varepsilon_{\text{target}}$, which is distinct from $\epsilon$. This separation ensures that the Rashomon set used for evaluation is controlled independently from the robustness tolerance $\epsilon$ used by ElliCE.

**Metrics.** We evaluate the generated counterfactual explanations based on four metrics: validity, proximity, robustness, and plausibility. *Validity* measures whether a generated counterfactual $\mathbf{x}_c$ for a given input $\mathbf{x}_0$ successfully achieves the desired outcome $c$ when evaluated on the original model $f_{\text{baseline}}$ for which it was generated, Validity $= \frac{1}{n}\sum_{i=1}^{n}\mathbf{1}[f_{\text{baseline}}(\mathbf{x}_{ci}) = c]$. *Proximity* measures the closeness of a counterfactual $\mathbf{x}_c$ to the original instance $\mathbf{x}_0$. We primarily report the $\ell_2$ distance: $\|\mathbf{x}_c - \mathbf{x}_0\|_2$. Lower values indicate less change required and are thus better. *Plausibility* checks whether the generated counterfactuals lie in realistic regions of the feature space. Our data-supported counterfactuals are inherently plausible, as they lie on the data manifold. For con-

tinuous approach, because ElliCE enforces robustness by pushing counterfactuals away from the decision boundary, the resulting counterfactuals might shift toward higher-density regions of the target class. Nevertheless, we evaluate plausibility using the Local Outlier Factor (LOF) [32], a standard outlier-detection metric. LOF values close to 1 indicate high plausibility, whereas larger values suggest the counterfactual is in a low-density region. *Robustness* computes whether the generated counterfactual $\mathbf{x}_c$ remains valid (i.e., still achieves the desired outcome $c$) for all models within an evaluator ensemble $\tilde{\mathcal{R}}(\varepsilon_{\text{target}})$. Total is calculated as the average across all $n$ counterfactual points: Robustness $= \frac{1}{n}\sum_{i=1}^{n}\mathbf{1}\left[\forall f_{\boldsymbol{\theta}}\in\tilde{\mathcal{R}}(\varepsilon_{\text{target}}), f_{\boldsymbol{\theta}}(\mathbf{x}_{c_i})=c\right]$. A higher robustness score (closer to 1) is better, indicating that more counterfactual explanations are robust to model changes.

**Experimental Setup.** For evaluators, we define a target multiplicity tolerance globally in range $\varepsilon_{\text{target}}\in[0,0.1]$. We provide discussion on how to choose ElliCE's $\epsilon$ in Appendix B. For every dataset, we performed 4-fold stratified cross-validation. Within each fold, the training data are further split into 80% for training and 20% for validation. The procedure within each inner fold is as follows: (1) We train a base model $f_{\text{baseline}}$, which serves as a reference model for all counterfactual generation methods. (2) Using $f_{\text{baseline}}$ as a reference (if required by the evaluation method), we generate $\varepsilon_{\text{target}}$-Rashomon set. (3) Multiplicity parameters for each baseline ($\epsilon$ for ElliCE, $\delta$ for Delta Robustness, ROAR and PROPLACE, or $\tau$ for T:Rex) are tuned via grid search on the validation set with a goal of maximizing validity. We allocate approximately the same amount of time for each method to tune its parameters with a hard maximum of 8 hours per method per data fold (as a result, we could not run PROPLACE for Parkinsons dataset). (4) Final performance metrics are reported on the held-out split of the outer fold. Note that due to our tuning procedure, we expect high validity metric for ElliCE and baselines. Indeed, for data-supported methods validity is consistently 100% across datasets, so we do not report it.

We conducted experiments on logistic regression and multilayer perceptrons. Consistent with prior work [32, 61], we focus on generating counterfactuals that change predicted labels from 0 to 1. Linear models are trained using Scikit-learn's LBFGS solver with an $\ell_2$ penalty (regularization parameter 0.001). MLPs are trained with the Adam optimizer (learning rate 0.001), early stopping, and $\ell_2$ regularization parameter 0.001. For evaluation, we generate one counterfactual per method for each data point in the held-out set. Each counterfactual is then evaluated against the three evaluators (Random Retrain, Rashomon Dropout, AWP). The exact construction algorithms for these evaluators are described in Appendix B.2. Reported metrics are averaged across data points and folds, with plots displaying the mean and standard error.

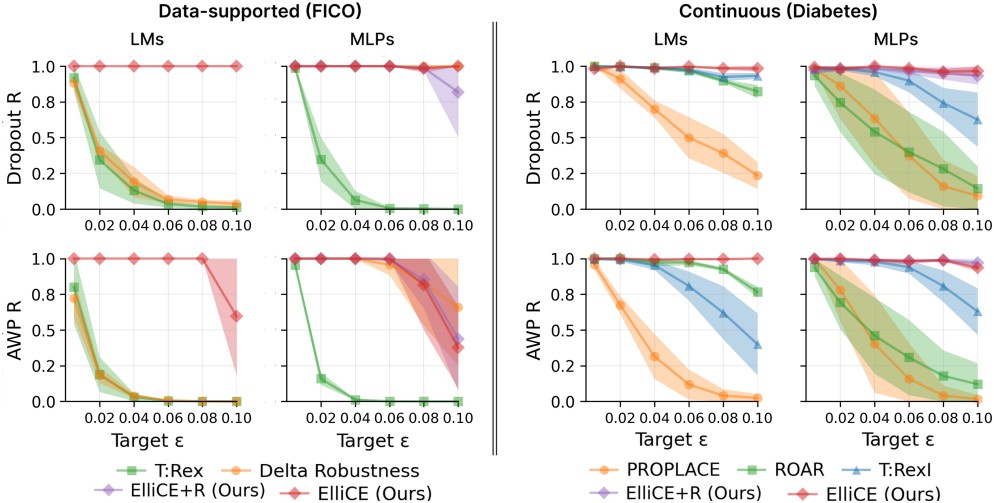

Figure 1: Robustness evaluation of ElliCE against baselines. The plot displays the robustness metric (y-axis) as a function of the target robustness level $\varepsilon_{\text{target}}$ for the evaluators (x-axis). ElliCE consistently outperforms all baselines across all robustness levels. See Appendix B for more figures. For ElliCE+R for MLPs, we apply additional regularization to the Hessian, using $\lambda = 0.1$ instead of 0.001.

Table 1: Performance of counterfactual methods on MLPs. For evaluators, we set $\varepsilon_{\text{target}}$ to 10% of the training objective ($\varepsilon_{\text{target}} = 0.1 \times \hat{L}(f_{\text{baseline}})$). **R** here stands for Robustness, **L2** for proximity, and PROP stands for PROPLACE. See Appendix B for results on other datasets.

| Data | Method | Evaluation Metric | | | | | |
| --- | --- | --- | --- | --- | --- | --- | --- |
| | | Retrain | | Dropout Rashomon | | AWP | |
| | | **R**↑ | **L2**↓ | **R**↑ | **L2**↓ | **R**↑ | **L2**↓ |
| | | Data-supported (DS) | | | | | |
| FICO | ElliCE | **1.00 ± 0.00** | 3.53 ± 0.17 | **1.00 ± 0.00** | 4.91 ± 0.22 | **1.00 ± 0.00** | 5.06 ± 0.29 |
| | DeltaRob | **1.00 ± 0.00** | 4.00 ± 0.10 | **1.00 ± 0.00** | 5.67 ± 0.58 | 0.96 ± 0.07 | 5.70 ± 0.72 |
| | T:Rex | 0.83 ± 0.08 | 3.12 ± 0.07 | 0.01 ± 0.00 | 3.07 ± 0.11 | 0.00 ± 0.00 | 2.77 ± 0.19 |
| German | ElliCE | **1.00 ± 0.00** | 3.48 ± 0.10 | **1.00 ± 0.00** | 4.32 ± 0.31 | **1.00 ± 0.00** | 4.00 ± 0.24 |
| | DeltaRob | 0.98 ± 0.01 | 3.45 ± 0.06 | 0.99 ± 0.02 | 4.00 ± 0.15 | **1.00 ± 0.00** | 3.99 ± 0.22 |
| | T:Rex | 0.99 ± 0.01 | 3.47 ± 0.04 | 0.97 ± 0.02 | 4.03 ± 0.20 | 0.99 ± 0.01 | 4.23 ± 0.24 |
| | | Continuous (CNT) | | | | | |
| Diabetes | ElliCE | **0.98 ± 0.01** | 2.15 ± 0.39 | **0.99 ± 0.02** | 3.05 ± 0.34 | **0.98 ± 0.02** | 3.22 ± 0.40 |
| | PROP | 0.48 ± 0.48 | 2.01 ± 0.05 | 0.19 ± 0.28 | 2.01 ± 0.05 | 0.08 ± 0.19 | 2.01 ± 0.05 |
| | ROAR | 0.86 ± 0.11 | 1.86 ± 0.24 | 0.40 ± 0.28 | 1.86 ± 0.24 | 0.31 ± 0.26 | 1.86 ± 0.24 |
| | T:Rex | 0.94 ± 0.03 | 2.47 ± 0.86 | 0.90 ± 0.08 | 4.18 ± 0.36 | 0.94 ± 0.04 | 4.18 ± 0.36 |

Table 2: Runtime performance and speedups for data-supported CE for MLPs.

| Dataset | Absolute (seconds) | | | Relative (speedup) | |
| --- | --- | --- | --- | --- | --- |
| | **ElliCE** | **T:Rex** | **Delta Rob** | **Over T:Rex** | **Over Delta Rob** |
| FICO | 1.792 ± 0.123 | 7.006 ± 0.058 | 242.035 ± 1.161 | 3.91× | 135.04× |
| COMPAS | 0.526 ± 0.011 | 3.534 ± 0.128 | 360.480 ± 6.701 | 6.72× | 685.34× |
| Australian | 0.057 ± 0.011 | 0.281 ± 0.006 | 2.783 ± 0.032 | 4.92× | 48.64× |
| Diabetes | 0.053 ± 0.001 | 0.296 ± 0.006 | 1.922 ± 0.032 | 5.60× | 36.33× |
| German | 0.101 ± 0.001 | 0.432 ± 0.013 | 9.905 ± 0.068 | 4.27× | 97.88× |

## 6.1 ElliCE Generates Robust Counterfactuals

Figure 1 illustrates the relationship between the evaluators' multiplicity level $\varepsilon_{\text{target}}$ and the achieved robustness for the baselines. We report results for both linear models and MLPs for data-supported and continuous methods. Across different settings, we observe that ElliCE consistently produces more robust counterfactuals than baselines. Notably, ElliCE's counterfactuals generally do not exhibit a decrease in robustness as $\varepsilon_{\text{target}}$ increases, demonstrating stability under different levels of target multiplicity. This robustness, however, can sometimes come with a greater distance from the original instance (i.e., longer CEs), a trade-off that we saw in Section 5 and report in Table 1. For the MLP setting, our empirical results in Figure 1 and Table 1 suggest that ElliCE's ellipsoidal approximation offers good flexibility, allowing it to adapt to the underlying loss function's shape.

## 6.2 ElliCE is Efficient

Tables 2, 5 and 6 clearly demonstrate ElliCE's advantage in computational efficiency. Our method is consistently faster than baselines with speedups of up to three orders of magnitude. The runtimes of both T:Rex and Delta Robustness tend to grow substantially with the dataset size. In contrast, ElliCE remains lightweight and exhibits better scalability. Across all datasets tested, ElliCE's absolute runtimes for generating a counterfactual remain under two seconds. This efficiency comes from a closed-form solution for the inner optimization problem (Theorem 1). The primary preprocessing cost involves computing and inverting the Hessian matrix $H$, requiring $O(np^2)$ for computation and $O(p^3)$ for inversion, performed once per model (where $n$ is the training set size and $p$ is the parameter dimension). Per-instance counterfactual generation then requires only $O(p^2)$ operations.

## 6.3 Sensitivity Analysis

Figure 2 (a,b) shows an empirical sensitivity analysis of ElliCE's robustness with respect to its internal Rashomon parameter $\epsilon$. The plots show how the achieved robustness (evaluated against the Random Retrain and Ellipsoidal Rashomon set evaluators, respectively) varies as ElliCE's internal $\epsilon$ changes. These results illustrate that ElliCE can achieve high levels of robustness even for rel-

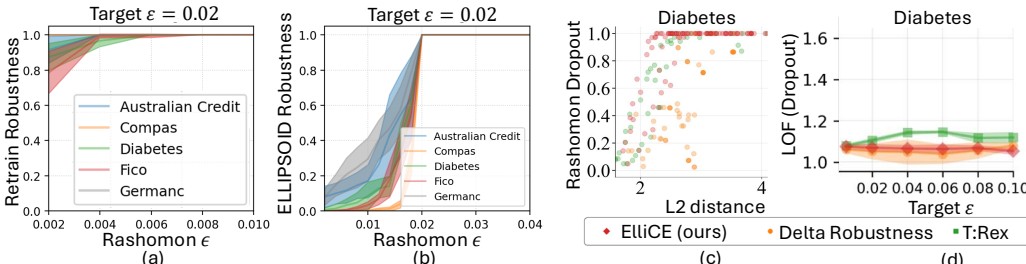

Figure 2: (a,b) Sensitivity of ElliCE's robustness (y-axis) to its internal $\epsilon$ hyperparameter (x-axis). Robustness is evaluated against Random Retrain (a) and an Ellipsoidal Rashomon set approximation defined with a fixed $\varepsilon_{\text{target}}$ (b). (c, d) Robustness vs. $\ell_2$ proximity trade-off (c) and plausibility (d) of counterfactuals generated by ElliCE and baselines on Diabetes dataset.

atively small values of its internal $\epsilon$ when evaluated against the Retrain ensemble. For the middle plot (Ellipsoidal evaluator), while initial robustness may be lower for smaller internal $\epsilon$ values, the performance increases sharply, as $\epsilon$ approaches the targeted robustness level.

### 6.4 Robustness-Proximity Trade-off and Plausibility

Figure 2(c) illustrates the inherent trade-off between robustness and proximity for CEs generated by ElliCE, supporting our discussion in Section 5. While the trade-off occurs for all baselines, ElliCE achieves the highest robustness at a given length level. Understanding this trade-off is key to selecting counterfactuals that balance reliability under model shifts with practical user actionability. ElliCE provides a mechanism to navigate this by allowing control over its Rashomon parameter. We also observed good plausibility across all baselines and datasets, as supported by Figure 2(d) and 8. All LOF values tend to be close to 1, thus the generated counterfactuals lie on the data manifold.

### 6.5 Actionability

To ensure that generated recourse remains realistic and feasible, we incorporate actionability constraints that specify which features can change and within what ranges. ElliCE supports restrictions on features, including immutable features (e.g., age, citizenship) as well as range and direction constraints such as income or loan duration. It also allows for sparse counterfactuals by adding an optional penalty on the number of modified features. For instance, before applying actionability, one robust counterfactual on the German Credit dataset suggested changing the applicant's age, an immutable feature. After enforcing immutability and sparsity constraints, ElliCE instead adjusted the credit amount and credit length, reducing both and thus lowering the predicted credit risk, which is reasonable in the lending context. Further details are provided in Appendix D.

## 7 Conclusions, Implications and Limitations

Standard algorithmic recourse is fragile. A recommendation given to a user today may become invalid tomorrow if the underlying model is retrained or replaced—a common scenario under the Rashomon Effect. This paper addressed this reliability gap by introducing ElliCE, a framework that provides recourse with provable robustness guarantees. ElliCE approximates the set of near-optimal models with an ellipsoid and computes counterfactuals that remain valid across this approximated Rashomon set. A strength of ElliCE is its support for actionability. Users can specify immutable features, range or direction constraints, and optional sparsity penalties, ensuring that the resulting recourse is both robust and realistic. This flexibility might help prevent impractical or unethical recommendations and gives users greater control over actions. While robustness alone does not ensure fairness, user-specified actionability constraints can help to ensure that counterfactuals remain feasible and ethically sound. A comprehensive fairness analysis remains an important direction for future work. The ellipsoidal approximation, while efficient, is a simplification of the true Rashomon set, and for neural networks our analysis currently captures local rather than global model multiplicity. Despite these limitations, ElliCE offers a practical and theoretically grounded tool for robust and actionable recourse, providing stable and trustworthy advice.

## Acknowledgments

We thank the RAI for Ukraine program, led by the Center for Responsible AI at New York University in collaboration with Ukrainian Catholic University in Lviv, for supporting Bohdan's and Iryna's participation in this research.

## Code Availability

Implementations of ElliCE are available at `https://github.com/BogdanTurbal/ElliCE_EXPERIMENTS`.

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
