# OpenReview forum: "ElliCE: Efficient and Provably Robust Algorithmic Recourse via the Rashomon Sets"
_NeurIPS.cc/2025/Conference — NeurIPS 2025 spotlight_

### Official Review · Reviewer_cC5H · 2025-06-30

**Clarity:** 2
**Significance:** 3
**Originality:** 3
**Rating:** 4
**Confidence:** 3

**Summary:**

This paper proposes a method that computes counterfactual explanations (CEs) that are robust to changes in the model if the changed model’s performance is within some bound of the original model. This is achieved by computing an ellipsoidal approximation of the Rashomon set (the set of all near-optimal models). This creates a quadratically constrained quadratic program that is solved as a second-order cone program. This formulation guarantees robustness to models in the Rashomon set (defined by some distance epsilon from the optimal solution), and also guarantees uniqueness and stability of the CE. The experiment results show that the proposed method is more robust than baselines, and significantly faster.

**Questions:**

1. Line 148, for MLPs, is the CE given in terms of the original feature space, or of the output values of the last MLP layer?
2. Line  154, is this approximation an over or under approximation?
3. Line 159, 160, how likely is the assumption that the Hessian matrix H is positive definite?
4. Line 198 states that the data-supported CE generation is used in section 6. Is the non-data supported CE generation also empirically tested? Or if not, why is it included in the text?
5. Line 251-254 mentions the trade-off between robustness and proximity of the CE. How, in practice, is the right value for epsilon chosen?
6. Why are two different target epsilon values used in Fig. 2? How are these values chosen?
7. Line 308-309, why is validity maximized in the grid search? Why not robustness or proximity or a trade-off?
8. Line 308-309, how can validity be used in the grid search for ElliCE if by line 206-209 validity is guaranteed, and line 322-323 also confirm 100% validity? Fig. 1, why is ElliCE’s robustness not always 100%?
9. Line 320-321, why does ElliCE operate on the features of the penultimate layer of a neural network? What guarantees does this provide?
10. How are the datasets presented in Table 1 selected? And why are robustness results in Table 4 much lower for the baselines?

**Ethical Concerns:**

["NO or VERY MINOR ethics concerns only"]

**Final Justification:**

My original score (2) was mainly based on (a) the paper's quality and clarity of writing, and (b) the amount of questions were triggered, specifically in the empirical section, which both did not reflect a high-quality submission.
Despite this, I actually agree with the other reviewers that the paper is well-written, as in comprehensible, but somewhat sloppy, and lacking important details.

The strength of the paper is that it proposes a strong method with useful theoretical and empirical analysis. If the current version of the paper gets rejected, I am sure that a future improved version will find publication.

Most of my questions have been answered well in the rebuttal. Specifically, my concern about the assumptions on the Hessian and on how the experiments were run are solved.

The question remains if the authors will succeed in improving the clarity or whether an updated version requires another round of review.

Given the strength of the method, and the other reviewer's positive appreciation of the work I ended up at a borderline accept.

**Limitations:**

The theoretical guarantees assume that the Hessian is positive definite. This is mentioned, but not reflected on how restrictive this assumption is.

**Quality:**

1

**Strengths And Weaknesses:**

Strengths:
- The proposed method improves on baselines in terms of runtime, robustness, and proximity of the explanation.

Weaknesses:
- The motivation for robustness under the Rashomon effect is weak
- The paper contains many notation and typographic mistakes, see the list below.
- The paper leaves too many open questions (see below)

Small comments:
- Missing space on line 73 after Perturbations.
- In line 98, the parameter vector is defined as an element of the (strict) subset of R^d, but if the hypothesis space is a multi-layer perceptron, this is not correct. Also, why is this Theta a strict subset?
- Line 116, R(epsilon) has a hat, while it was defined without one in line 109. Also line 296, or is this the approximated Rashomon set defined in line 278-289?
- Line 120, “an data point”
- Appendix ?? several times
- Line 134, the full stop between l_mix and l_2 can easily be mistaken for a multiplication dot.
- Line 138, why switch from x_c and x_0 to x and x’? Similarly, line 139 switches to accessing x by one index i (or j), whereas line 136 uses two indices. This is inconsistent.
- Line 135 defines v_l_2, whereas  line 138 defines l_mix. Why not define either both v_l_2 and v_l_mix, or both l_2 and l_mix?
- Line 151 uses p for the number of features, while this d is used in line 95, and even the very next line 152 uses d also. Line 225 uses p again.
- Line 151 missing a closing bracket } before ([44]) also, why () around [44]?
- Line 216, x_c is not bold (twice)
- Line 293, double usage of the symbol c
- Line 293, switched from using the big one notation (line 139-140) to big I for the indicator function. Line 298 switches back again.
- The fonts in Fig. 1 are too small
- Table 1 and Fig. 1 appear two pages before they are mentioned in the text
- Table 1, R is not defined. Probably the robustness?
- Fig. 2 ROB robustness is not defined
- Figures use a different epsilon symbol than the main text (\epsilon vs \varepsilon)
- Links in the appendix don’t work
- Table 3 in the appendix extends into the margins
- Table 4 in the appendix does not describe what the dash means
- Table 4 does not consistently mark lowest proximity values bold

---

> ### Author Rebuttal · Authors · 2025-07-31
>
> Thank you so much for your review and feedback.
>
> > **W1: The motivation for robustness under the Rashomon effect is weak.**
>
> The robustness under the Rashomon effect is a critical and practical motivation for our work. Most counterfactual explanation methods operate on the assumption that the underlying predictive model is fixed. They generate explanations, often very close to the decision boundary, that are valid for one specific model. In practice, this assumption rarely holds. In many real-world applications, especially on tabular data, numerous distinct models can achieve similar, near-optimal performance. This is known as the Rashomon effect, or model multiplicity. Institutions may switch between these equally good models as part of regular maintenance, retraining cycles, or to adapt to minor shifts in data. This leads to a significant reliability problem. A counterfactual action guaranteed to work for one model may fail completely if another one is deployed. This undermines the entire purpose of algorithmic recourse, which is to provide reliable and actionable paths to a desired outcome.
>
> Our work, ElliCE, directly confronts this failure point. Instead of generating an explanation for a single model, we produce recourse that is provably valid across an entire ellipsoidal approximation of the Rashomon set. This ensures the explanation given to individuals is a robust and reliable action, closing a gap in providing trustworthy algorithmic recourse under real-world model uncertainty.
>
> > **Small comments**
>
> Thank you for such a detailed list of typos and suggestions. We will incorporate them, including fixing typos, notation, increasing fonts, improving margins, and so on. Next, we focus on clarifying questions and details.
>
> > **In line 98, the parameter vector is defined as an element of the (strict) subset of R^d, but if the hypothesis space is a multi-layer perceptron, this is not correct. Also, why is this Theta a strict subset?**
>
> The notation is a general formalization. For MLP, d represents the total number of weights and biases across all layers. The strictness is not necessary. It is a common formality used for the convenience of theoretical results.
>
> > **Line 116, R(epsilon) has a hat, while it was defined without one in line 109. Also line 296, or is this the approximated Rashomon set defined in line 278-289?**
>
> We used a hat to show that the Rashomon set is empirical, as constructed on the empirical data and not on the distribution. We will improve the consistency of the notation.
>
> > **Line 148, for MLPs, is the CE given in terms of the original feature space, or of the output values of the last MLP layer?**
>
> The counterfactual explanations are given in the original feature space. While we construct the ellipsoidal approximation of the Rashomon set in the embedding space produced by the penultimate layer of the neural network, we map this information back to the original feature space. Our algorithms (both gradient-based and cutting plane methods) operate directly in the input space while using the worst-case model identification from the embedding space to guide the search for robust counterfactuals. This approach allows us to provide explanations in terms of the original features that users can understand and act upon.
>
> > **Line 154, is this approximation an over or under approximation?**
>
> Using the Hessian matrix to define the set generally results in an under-approximation of the Rashomon set. The ellipsoidal set can be understood as a tractable geometric approximation based on the local curvature of the loss function.
>
> Please note that for L2 regularized mean-squared error, the Rashomon set can be computed exactly as an ellipsoid, meaning it is not an approximation in that scenario.
>
> > **Limitations and Line 159, 160, how likely is the assumption that the Hessian matrix H is positive definite?**
>
> The positive definiteness of the Hessian matrix is a reasonable assumption in our context. For linear models with convex losses, the Hessian is positive definite by definition. For neural networks, where the loss landscape may be non-convex, we ensure numerical stability by adding a small regularization term to the computed Hessian (equivalent to L2 regularization with a parameter). This practice is standard in optimization literature and guarantees that our ellipsoidal approximation remains well-defined. Our experiments confirm that this approach works effectively across diverse datasets and different model classes. Therefore, the positive definite Hessian assumption is a standard condition in the context of regularized models and is satisfied within our paper’s framework.
>
> > **Line 198 states that the data-supported CE generation is used in section 6. Is the non-data supported CE generation also empirically tested? Or if not, why is it included in the text?**
>
> Yes, we provide the results for non-data supported counterfactual generation in Appendix C.7. We couldn’t include these results in the main paper due to space constraints, so we moved a lot of empirical findings to Appendix.
>
> > **Line 251-254 mentions the trade-off between robustness and proximity of the CE. How, in practice, is the right value for epsilon chosen?**
>
> As we state in Section 6, all multiplicity parameters ($\epsilon$ for ElliCE, $\delta$ for Delta Robustness, or $\tau$ for T:Rex)  for each baseline are tuned via grid search on the validation set, basically maximizing validity. Given the robustness-proximity trade-off, this approach ensures that the counterfactual explanations are long enough to ensure the basic requirement, which is validity, but still satisfy the proximity requirement. We use grid search to provide each method the capacity to be robust, as the parameters of the baselines are not interpretable, and the papers of the baseline method often do not provide details on how to choose them. We provide guidelines on selecting a suitable epsilon in Appendix C4
>
> > **Why are two different target epsilon values used in Fig. 2? How are these values chosen?**
>
> The target epsilon defines the allowable tolerance in empirical loss used to construct the evaluator baselines (Random Retrain, AWP, and Rashomon Dropout). For the analyses in Figure 2, the values of 0.02 and 0.04 were chosen arbitrarily from within a predefined range of [0, 0.1] as representative examples of possible loss increases due to model multiplicity.
>
> > **Line 308-309, why is validity maximized in the grid search? Why not robustness or proximity or a trade-off?**
>
> We maximize validity during grid search because it's the fundamental prerequisite for counterfactual explanations. Optimizing directly for robustness would create circular reasoning since our evaluators depend on the same parameters we're tuning. Validity provides a fair comparison baseline across methods, after which the inherent robustness-proximity trade-offs of each method can be meaningfully compared.
>
> > **Line 308-309, how can validity be used in the grid search for ElliCE if by line 206-209 validity is guaranteed, and line 322-323 also confirm 100% validity? Fig. 1, why is ElliCE’s robustness not always 100%?**
>
> For a given data point, a valid counterfactual might not exist if the robustness constraint ($\epsilon$) is too large, making the problem infeasible. The grid search tunes epsilon to maximize the number of data points for which a valid counterfactual can be found.
> ElliCE's robustness is not always 100% because it is guaranteed for its internal ellipsoidal approximation, but it is evaluated against different evaluators (e.g., Random Retrain, AWP, Dripout) . Since these approximation sets are not identical, a counterfactual proven robust for one set may not be valid for all models in the other.
>
> > **Line 320-321, why does ElliCE operate on the features of the penultimate layer of a neural network? What guarantees does this provide?**
>
> ElliCE operates on the penultimate layer to treat the neural network's final layer as a simple linear model. This makes the problem tractable and allows the same convex optimization framework to be applied directly.
>
> > **How are the datasets presented in Table 1 selected? And why are robustness results in Table 4 much lower for the baselines?**
>
> Table 1 or the main text is a subtable of Table 4 from the Appendix. The choice of datasets is random. The robustness results for the baselines are lower because ElliCE performs better.

---

> > ### Comment · Reviewer_cC5H · 2025-08-04
> >
> > Thank you for your detailed responses. I believe the paper proposes a strong method with useful theoretical and empirical analysis. My main concern as above is the paper's quality and clarity. Even though they can be easily fixed, I would expect a submission to NeurIPS not to have so many mistakes in notation. My questions have been answered well, and I hope incorporating these answers in the text will make the text more clear, also for people less knowledgeable in the field. Specifically my concern about the assumptions for the Hessian, and on how the experiments were run are solved. Therefore, I want to raise my score under the assumption that these clarifications will find a way in the updated text.

---

> > > ### Author Response · Authors · 2025-08-05
> > >
> > > Thank you for recognizing the contributions of our theoretical and empirical analysis. We appreciate the detailed list of suggestions and typos, and we will address them to ensure the notation and overall presentation are consistent and accessible. We also sincerely appreciate your willingness to raise the rating.

---

### Official Review · Reviewer_MTx2 · 2025-07-01

**Clarity:** 4
**Significance:** 3
**Originality:** 3
**Rating:** 5
**Confidence:** 5

**Summary:**

The paper proposes a new method to generate robust counterfactual explanations targeting linear models and MLP for binary classification. This is done by modelling the rashomon set as an ellipsoid in the weight space and formulating an optimization problem for finding robust and closest counterfactuals. Theoretical results on robustness and stability are given for linear models. Experiments involved many tabular datasets with linear models and MLPs. Quality of counterfactuals is evaluated using robustness and proximity metrics, and experiments show better results in these two aspects than two strong baselines.

**Questions:**

1. Can you comment on the plausibility performance of the proposed method?
2. Does the proposed method also apply beyond the binary classification setting?
3. It appears in the code that baseline implementations are taken from the ROBUSTX library, which is not mentioned in the paper.

Please also see weakness section above.

**Ethical Concerns:**

["NO or VERY MINOR ethics concerns only"]

**Final Justification:**

The paper proposes a new method that targets the robustness of counterfactual explanations for recourse purposes. Only a few works in the literature provide provable robustness guarantees in the literature, and this work has it, which is very nice. The biggest concern I had was the lack of detailed quantitative evaluations for the plausibility aspect, and this was addressed in the rebuttal, along with some other minor concerns. I hence raise my rating to 5 and recommend accept.

**Limitations:**

Limitations are discussed in the paper.

**Quality:**

3

**Strengths And Weaknesses:**

Strengths:
1. The paper is very well-written. Motivations and problem settings are clear, mathematical definitions are easy to follow, and assumptions are clearly stated.
2. The proposed method makes sense and works well for the defined scope (robustness for rashomon set + proximity + time). The accompanying theoretical results provide clear evidence of why it works. The way to represent Rashomon set with an ellipsoid is novel.
3. A good number of baselines and datasets are involved in the experiments (incl. Appendix). Diverse forms of rashomon set construction are included too.

Weakness:
1. Quality and Significance: The **main concern I have** is the fact that there is no quantitative evaluation about plausibility (data manifold). The two baselines for the evaluation section (RNCE and TREX) are both designed for robustness, proximity, AND plausibility. It is known that there is a proximity-plausibility trade-off for counterfactuals, indicating that both baselines might not have the best proximity in order to achieve plausibility. Therefore, in order to show that the proposed method is indeed better than those two, I think better, or at least comparable, quantitative results in plausibility are also essential, in addition to evaluating robustness and proximity.
2. Significance: The method applies only to the linear classifier case (line 98 $\theta \in \mathbb{R}^d$) and to binary classification. This is a minor inherent limitation since some previous published works also target such scenarios.
3. Significance and originality: while the ellipsoid idea is novel (strength No.2), the optimization approach in line 167 and Equation 1 which is the main part of computing the counterfactual, has been quite standard in the literature. The same formulation was used in, e.g., reference [48], [23], and a similar formulation was used in Dominguez-Olmedo et al., 2022 ICML, and Maragno et al., 2023 INFORMS Journal on Computing.

---

> ### Author Rebuttal · Authors · 2025-07-31
>
> Thank you very much for your review and feedback.
>
> > **W1 and Q1: Can you comment on the plausibility performance of the proposed method?**
>
> Thank you for asking about plausibility. Our data-supported counterfactuals are inherently plausible, as they lie on the data manifold.
> As you mentioned, for non-data supported counterfactuals, there are possible cases when the resulting counterfactuals for continuous data may be in low-density (not plausible) areas. Very often, it happens when the resulting counterfactual is too close to the decision boundary. Note that due to our robustness requirements, we push counterfactuals away from the decision boundary. We can illustrate this on the synthetic examples, based on two Gaussian distributions, where each class $P(y|x)$ comes from a separate Gaussian. Visually, we observe that the resulting ElliCE counterfactuals are pushed towards the mean of the Gaussian of the target class. Visually, they also look in good density regions, where other points are also present. We will add this illustration on the synthetic data to the paper. Also, note that we observe a robustness-proximity trade-off, as we report in Theorem 5.
>
> Counterfactuals might not be plausible when we push too far away from the decision boundary. In case of ElliCE, its hyperparameter epsilon allows us to control for this, as we seek for closest examples for some robustness level.
>
> To provide more rigorous analysis, we evaluated the plausibility of the counterfactuals using LOF metric (as in [1]). LOF is used for outlier detection, and values close to 1 are preferred. Please see our results in the table below. For data-supported counterfactuals, we have small values comparable to baselines. For non-data supported counterfactuals, ElliCE performs slightly better than TREX and better than ROAR on the diabetes dataset. PROPLACE, which is designed for plausibility, achieves better results on LOF. However, please also note that ElliCE achieved better robustness results as compared to PROPLACE.
> We will add a more comprehensive study of the plausibility to the paper, if accepted.
>
> Table: Plausibility study of Ellice and baselines under different evaluation strategies.
> | **AWP**               | **Evaluator** |             |             |             |             |\|     |        |             |             |             |
> |-----------------------|:-------------:|:-----------:|:-----------:|:-----------:|:-----------:|-----|:--------------------:|:-----------:|:-----------:|:-----------:|
> | **Non Data-supported**|               |             |             |             |             |   \|    | **Data-supported**   |             |             |             |
> | **Dataset**           | **ElliCE**    | **PROPLACE**| **ROAR**    | **TRexI**   |            |  \|     | **Dataset**          | **ElliCE**  | **RNCE**    | **STCE**    |
> | Bank                  | 1.493±0.046   | 1.106±0.029 | 1.310±0.066 | 1.492±0.075 |             |   \|    | Austr                | 1.048±0.015 | 1.090±0.040 | 1.077±0.038 |
> | Diab                  | 1.197±0.045   | 1.076±0.027 | 1.249±0.113 | 1.558±0.098 |                |   \|    | Germ                 | 1.023±0.003 | 1.026±0.001 | 1.032±0.006 |
> | **ROB**               | **Evaluator** |             |             |             |             |    \|   |        |             |             |             |
> ||||||||||||
> | **Non Data-supported**|               |             |             |             |             |   \|    | **Data-supported**    |             |             |             |
> | **Dataset**           | **ElliCE**    | **PROPLACE**| **ROAR**    | **TRexI**   |             |  \|     | **Dataset**           | **ElliCE**  | **RNCE**    | **STCE**    |
> | Bank                  | 1.488±0.033   | 1.097±0.033 | 1.310±0.066 | 1.492±0.075 |             |  \|     | Austr                 | 1.039±0.016 | 1.082±0.040 | 1.067±0.036 |
> | Diab                  | 1.194±0.047   | 1.076±0.020 | 1.274±0.134 | 1.558±0.098 |             |  \|     | Germ                  | 1.024±0.005 | 1.027±0.002 | 1.031±0.006 |
> | **Retrain**           | **Evaluator** |             |             |             |             |   \|    |         |             |             |             |
> ||||||||||||
> | **Non Data-supported**|               |             |             |             |             | \|      | **Data-supported**    |             |             |             |
> | **Dataset**           | **ElliCE**    | **PROPLACE**| **ROAR**    | **TRexI**   |             |    \|   | **Dataset**           | **ElliCE**  | **RNCE**    | **STCE**    |
> | Bank                  | 1.450±0.045   | 1.136±0.023 | 1.319±0.074 | 1.478±0.071 |             |   \|    | Austr                 | 1.052±0.013 | 1.057±0.015 | 1.051±0.016 |
> | Diab                  | 1.145±0.042   | 1.039±0.014 | 1.338±0.092 | 1.207±0.050 |             |  \|     | Germ                  | 1.024±0.006 | 1.017±0.005 | 1.020±0.003 |
>
> To further help to control counterfactuals' actionability and plausibility, we did a series of updates to our algorithm. Besides having immutable features, we now allow:
>
> 1) Change of the feature in one direction (as if the value can only increase but not decrease, say if a person plans to increase income or spending amount when applying for a credit). This is implemented by blocking gradient components that would move features in the prohibited direction.  For the data-supported method we do prefiltering.
>
> 2) Fixed range for the feature. This is achieved by restricting gradient flow to stay within bounds and pruning KD-tree searches to exclude out-of-range solutions.
>
> 3) sparsity of the counterfactual explanations. We additionally employ L0 and L1 regularizations to allow for more sparse explanations for the cases when the user wants to change a small number of features.  For data-supported counterfactuals, this is implemented by defining a custom distance metric for Ball tree search that combines sparsity and proximity ($metric = C \times \ell_0 + \ell_1$, where a coefficient C prioritizes minimizing the number of changed features).
> We hope that these changes will help the users to find more plausible and actionable counterfactuals and overall contribute to the flexibility of our framework.
>
> [1] Jiang, Junqi, et al. "Provably robust and plausible counterfactual explanations for neural networks via robust optimisation." Asian Conference on Machine Learning. PMLR, 2024.
>
> > **W2: The method applies only to the linear classifier case (line 98 ) and to binary classification.**
>
> ElliCE works beyond linear models and works for non-linear hypothesis spaces that can be optimized with gradient descent.
> In our experimental section, we included results for the multi-layer perceptrons and show that our algorithm effectively works for this case. We describe in the Appendix in Algorithms 1 and 2 the method that we use to compute counterfactuals for the neural network space. Our theoretical section applies to the penultimate layer of the MLP. In this layer, we do the Rashomon set approximation and use our theory in Theorem 1 to compute the worst-case model for the Rashomon set.
> Please note that a lot of counterfactual generation methods known to us focus on binary classification (except PROPLACE), as it is very popular in tabular data settings. However, it is easy to extend ElliCE to the multiclass setting. We describe this extension below.
>
> > **Q2: Does the proposed method also apply beyond the binary classification setting?**
>
> There is a simple way to extend our framework to handle one-versus-one multiclass classification. For one versus one, we will assume that we know the target class for the counterfactual. Our task succeeded if, for the resulting counterfactual, the target class’s $y$ logit is higher than each logit of some other class $c$ individually, under the worst-case model within the Rashomon set.
>
> Formally, for a counterfactual point $x_c$:
> $$
> \min_{\hat{\mathcal{R}}(\epsilon)} (f_{\theta}^{y} (x_c) - f_{\theta}^{c} (x_c)) \ge \tau, \qquad \forall \space c \neq y
> $$
>
> where $\hat{\mathcal{R}}(\epsilon)$ is our ellipsoidal approximation of the Rashomon set, and $\tau$ is a margin hyperparameter that we can set to 0. This guarantees that the explanation remains valid across all models within the Rashomon set.
>
> This way, we get $K−1$ second-order constraints (one for each rival class) and can use the same closed-form inner optimization as in the binary case. The overall optimization problem remains a convex second-order cone program (SOCP). The runtime grows roughly linearly in $K$.
>
> For neural networks, we apply the same approach using a Rashomon set approximation in the embedding space and a gradient descent or cutting-plane approach, as in our binary case.
>
> > **Q3: It appears in the code that baseline implementations are taken from the ROBUSTX library, which is not mentioned in the paper.**
>
> Thank you for pointing this out. You are correct, we used baseline implementations from the ROBUSTX library [2]. We will update the manuscript to include a proper citation and clarify that baselines implementation was done using ROBUSTX.
>
> [2] Jiang, Junqi, et al. "RobustX: Robust Counterfactual Explanations Made Easy." arXiv preprint arXiv:2502.13751 (2025).
>
> > **W3: The optimization approach in line 167 and Equation 1 is standard.**
>
> We actually think it is a good thing that the definition and high-level optimization problem for the counterfactual explanation is standardized in the literature and community. Note that the optimization problems that you mentioned are quite general: minimize the distance with respect to some constraints. And here is where ElliCE shines, as we provide new way of handling constraints: using the Rashomon set that we approximate with an ellipse. Our experiments show that this approximation of the Rashomon set is effective when we use multiplicity evaluation baselines such as AWP and random retrain.

---

> > ### Comment · Reviewer_MTx2 · 2025-08-05
> > **Response to author rebuttal**
> >
> > Thank you for the detailed response. The additional plausibility experiments using the local outlier factor score reasonably characterize the performance of the proposed method - comparable with SOTA for the data-supported version and comparable with methods not targeting plausibility for the non-data-supported version. Adding these will allow a fair and thorough comparison with the baselines. The extension to multi-class also makes sense.
> >
> > My concerns have been addressed. I am willing to raise my rating to 5, accept.

---

> > > ### Author Response · Authors · 2025-08-05
> > >
> > > Thank you for your thoughtful feedback and for helping improve the quality of our work. We will incorporate all the suggested changes to support a more inclusive evaluation and discussion. We also sincerely appreciate your willingness to raise the rating.

---

### Official Review · Reviewer_WWfa · 2025-07-01

**Clarity:** 4
**Significance:** 3
**Originality:** 3
**Rating:** 5
**Confidence:** 4

**Summary:**

The authors propose an algorithm for generating counterfactual explanations which are robust to model multiplicity. This is done by employing an ellipsoidal approximation for the Rashomon set, into which the considered model belongs. The trade-off between the Rashomon parameter $\epsilon$ and the cost of the generated counterfactual (L2 distance to be specific) is analytically derived in case of linear models. Other important properties of the generated counterfactuals such as the uniqueness and monotonicity of the cost w.r.t. the Rashomon parameter are theoretically established. Experiments, conducted over five real-world datasets, demonstrating the performance of the algorithm are presented.

**Questions:**

1. How does the proposed method compare with the baselines w.r.t. the robustness to changes in training data or re-training on initially unseen data?
2. Does Theorem 1 still hold in case of a non-linear models (with the assumption of local linearity or linearization in the final layer in case of MLPs)?

**Ethical Concerns:**

["NO or VERY MINOR ethics concerns only"]

**Final Justification:**

I thank the authors for the detailed response. The rebuttal has addressed all my concerns. I believe that it is extremely important to highlight the correspondence of the Rashomon set and the set of models robust to different types of naturally-occurring multiplicities. The additional experiments done by the authors showcase this correspondence, and hence I am happy to raise the rating. I also acknowledge the proposed revisions to the theorem statement and hope that it would be incorporated in the final version.

**Limitations:**

Yes

**Paper Formatting Concerns:**

No major formatting concerns. Minor comments:
1. Cited appendix numbers are missing in the main text.
2. Reference [51] and [52] looks the same.

**Quality:**

3

**Strengths And Weaknesses:**

*Strengths:*
1. The idea of quantifying robustness of counterfactual explanations w.r.t. an ellipsoidal approximation of the Rashomon set is novel and interesting. Moreover, the experiments demonstrate that this approximation works well enough for other scenarios where the theoretical guarantee does not directly apply.
2. Experiments are well-detailed and the results are clear. The proposed algorithm shows a significant improvement in counterfactual generating time compared to the baselines (not considering the Hessian and Hessian-inverse computations, which are done once for a training set and a model).
3. The paper is clearly written and well-organized.

*Weaknesses:*
1. The main weakness that I am concerned about is the suitability of this approximation of the Rashomon set for characterizing the kind of model multiplicity which can occur in the potential real-world applications. Most of the baselines considered in this paper (e.g.: Trex, Argumentative Ensembling) construct the multiplicity set by changing the training data--hence robustness to changes in training data and retraining on newly arrived data--which is not a scenario considered in this paper.

2. As the authors have already acknowledged, the theoretical analysis is only valid for linear models (or models which allow for such simplifications) which is very restrictive given the domain of potential applications of the algorithm. Moreover, although the authors have mentioned "Without loss of generality we focus our theoretical analysis on linear predictors..." in the beginning of Section 4, I do not see how the analysis applies to other non-linear models.

As I see, the above two factors largely impact the significance of the theoretical and empirical contributions of the paper.

*Minor comments on clarity:*
1. Cited appendix numbers are missing in the main text.
2. Reference [51] and [52] looks the same.

---

> ### Author Rebuttal · Authors · 2025-07-31
>
> > **W1: Multiplicity computing approach**
>
> The ellipsoid-based Rashomon sets will contain models that occur due to slight changes in the training data that T:Rex, Argumentative Ensembling are considering, especially for convex losses, such as logistic loss. This is because this Rashomon set contains different coefficient vectors (and thus feature weighting), allowing for diversity of the models. Therefore, if the data shift occurred, there are other models to choose from the ellipsoidal Rashomon set to account for the changes. To support this, we run a simple experiment to show the stability of the ellipsoid-based Rashomon set (see table below). More specifically, we add noise to the data that comes from a Gaussian distribution with 0 mean and variance $\sigma$ that ranges from 0.1 to 2.
>
> We consider the hypothesis space of linear models and the German Credit dataset. First, we compute the Rashomon set on the original data (we use 10% of the optimal loss as epsilon and standardize the data). For each variance value, for 100 times, we do the following: 1) sample noise, 2) create a new dataset with added noise, 3) train logistic regression on this new data, 4) check if the Rashomon set contains models with similar performance as this optimal model. Finally, we compute the percentage of cases when the Rashomon set contained a similar performing model on this new data as the optimal one in the table below. We find that the Rashomon set is stable, even for higher levels of noise. Therefore, we expect the ellipsoid to contain models that perform similarly to optimal models on data folds (if not directly containing these optimal models).
>
> Table. Fraction of cases when the Rashomon set trained on original data contains optimal models trained on noise data, with noise $\sim \mathcal{N}(0, \sigma)$.
> | **Applied Variance**, $\sigma$ | **% Models Inside Ellipsoid** |
> |----------------------|-------------------------------|
> | 0.1                  | 100.0%                        |
> | 0.2                  | 100.0%                        |
> | 0.3                  | 100.0%                        |
> | 0.4                  | 100.0%                        |
> | 0.5                  | 100.0%                        |
> | 0.6                  | 100.0%                        |
> | 0.7                  | 100.0%                        |
> | 0.8                  | 100.0%                        |
> | 0.9                  | 100.0%                        |
> | 1.0                  | 100.0%                        |
> | 1.1                  | 100.0%                        |
> | 1.2                  | 100.0%                        |
> | 1.3                  | 100.0%                        |
> | 1.4                  | 100.0%                        |
> | 1.5                  | 100.0%                        |
> | 1.6                  | 97.0%                         |
> | 1.7                  | 98.0%                         |
> | 1.8                  | 96.0%                         |
> | 1.9                  | 87.0%                         |
>
> > **Q1. How does the proposed method compare with the baselines w.r.t. the robustness to changes in training data or re-training on initially unseen data?**
>
> To further support that the Rashomon set handles re-training scenarios, we followed a similar setup that T:Rex and Argumentative Ensembling are using. More specifically, we split the data in 2 parts. We use first data part to train our Rashomon set (ElliCE) and train baselines (STCE, RNCE). Second data part we use to train evaluators (random retraining, adversarial weight perturbation, and Rashomon dropout). This, similar to T:Rex and Argumentative Ensembling, considers different data for the methods we evaluate and baselines with which we do the evaluation. For this new evaluation, we still found that ElliCE outperforms or performs on par with the baselines. Since we can not report images, please see the tables below. Here we use Target Epsilon 0.02.
>
> Table: Re-train evaluation of ElliCE for the hypothesis space of linear models
>
> | Dataset    | Method   | Retrain | ROB | AWP     |
> |-----------|-----------|-----------|---------|---------|
> | **COMPAS**     | ElliCE   | 1.0000 ± 0.0000          | 1.0000 ± 0.0000        | 1.0000 ± 0.0000 |
> |            | RNCE     | 1.0000 ± 0.0000          | 0.3377 ± 0.2214        | 0.1121 ± 0.033 |
> |            | STCE     | 0.9982 ± 0.0032          | 0.9994 ± 0.0010        | 1.0000 ± 0.0000 |
> ||||||
> | **AustrC**     | ElliCE   | 0.9604 ± 0.0495          | 1.0000 ± 0.0000        | 1.0000 ± 0.0000 |
> |            | RNCE     | 0.9074 ± 0.1604          | 0.7145 ± 0.2481        | 0.7305 ± 0.2320 |
> |            | STCE     | 0.9564 ± 0.0392          | 0.9030 ± 0.0891        | 0.9195 ± 0.0804 |
> ||||||
> | **GermanC**    | ElliCE   | 0.9938 ± 0.0108          | 0.9688 ± 0.0541        | 0.9688 ± 0.0541 |
> |            | RNCE     | 0.7880 ± 0.1982          | 0.6239 ± 0.2200        | 0.6614 ± 0.2082 |
> |            | STCE     | 0.7230 ± 0.1399          | 0.5727 ± 0.1416        | 0.6219 ± 0.1305 |
>
> Table: Re-train evaluation of ElliCE for the hypothesis space of multi-layer perceptrons.
> | Dataset    | Method   | Retrain | ROB | AWP     |
> |-----------|-----------|-----------|---------|---------|
> | **COMPAS**     | ElliCE   | 0.9994 ± 0.0010          | 0.9994 ± 0.0010        | 1.0000 ± 0.0000 |
> |            | RNCE     | 0.9994 ± 0.0010          | 0.9988 ± 0.0021        | 0.9994 ± 0.0010 |
> |            | STCE     | 0.9989 ± 0.0020          | 0.7522 ± 0.1958        | 0.9193 ± 0.1326 |
> ||||||
> | **AustrC**     | ElliCE   | 0.9305 ± 0.0696          | 0.9865 ± 0.0234        | 0.9527 ± 0.0819 |
> |            | RNCE     | 0.9643 ± 0.0619          | 0.9459 ± 0.0936        | 0.9459 ± 0.0936 |
> |            | STCE     | 0.9554 ± 0.0773          | 1.0000 ± 0.0000        | 1.0000 ± 0.0000 |
> ||||||
> | **GermanC**    | ElliCE   | 1.0000 ± 0.0000          | 0.9167 ± 0.1443        | 1.0000 ± 0.0000 |
> |            | RNCE     | 0.8910 ± 0.1887          | 0.9038 ± 0.1665        | 0.9038 ± 0.1665 |
> |            | STCE     | 1.0000 ± 0.0000          | 1.0000 ± 0.0000        | 1.0000 ± 0.0000 |
>
> > **W2 and Q2: The applicability of the results to the non-linear case.**
>
> Thank you for pointing this out. “Without loss of generality” in the paragraph on line 146 applies to the second sentence, as our results apply to the embedding space of the neural networks (penultimate layer). In this layer, we do the Rashomon set approximation and use our theory from Theorem 1 to compute the worst-case model for our Rashomon set. Then we can use 2 methods to find counterfactuals in the feature space:
> 1) gradient-descent method that is described in Appendix C7. On each step of the algorithm, we compute the weighted sum of the worst base model (based on the formula in Theorem and the optimal model, then do the gradient step
>
> 2) cutting-place method that is described on line 187.
>
> We found the cutting plane method to work very well and better than baselines for smaller Rashomon set epsilons. This method tends to converge faster than gradient descent for smaller epsilons as well. For larger epsilon, we found that the gradient-based method is more stable.
>
> Both of these methods work for neural networks and allow for robust counterfactuals as our results suggest in the experimental section and appendix. For both, we use our theoretical results from Section 4. So no, *our theory does not apply only to linear models*.
>
> We will explain the methods and how we use our theoretical results more clearly in the paper. We will also remove “Without loss of generality” at the start of line 146 to not confuse the readers, as it refers to the second sentence. Our new paragraph will say: “We focus our theoretical analysis on linear predictors of the form $f_{\theta}(x)=\theta^{\top}x$. However, the methodology presented here is also applicable to the embedding space of the multilayer perceptrons.”
>
> We will also fix minor formatting concerns. Thank you again for your review.

---

### Official Review · Reviewer_qf5X · 2025-07-02

**Clarity:** 2
**Significance:** 3
**Originality:** 3
**Rating:** 5
**Confidence:** 3

**Summary:**

The paper introduces **ElliCE**, a framework for generating robust counterfactual explanations under the Rashomon effect. By approximating the Rashomon set as an ellipsoid derived from loss landscape geometry, ElliCE frames robust counterfactual generation as a convex optimization problem. The method claims theoretical guarantees on uniqueness, stability, and alignment with feature importance directions, and empirically outperforms baselines in robustness and computational efficiency across diverse datasets.

**Questions:**

1. How does your method perform when the true Rashomon set has non-convex or disjoint regions like multiple local optima? For example, in cases where models cluster into distinct subspaces, would the ellipsoidal approximation catastrophically fail? Could this lead to overly conservative counterfactuals that sacrifice proximity for robustness?
2. The experiments focus on linear models and neural networks with final-layer linearization. How does ElliCE handle deeper architectures? Does linearizing intermediate layers compromise the geometric intuition of the Rashomon set, leading to less robust counterfactuals? What trade-offs arise in non-linear regimes?
3. Section 4 mentions "feature-level constraints". How are these constraints integrated into the optimization?
4. The method assumes a positive definite Hessian for closed-form solutions. What happens when this assumption fails? For example for non-convex losses or high-dimensional data? How is numerical stability maintained in such cases?

**Ethical Concerns:**

["NO or VERY MINOR ethics concerns only"]

**Final Justification:**

In the rebuttal discussion, the authors successfully resolved all concerns. I increased the score to 5, conditionally on the improved writing in the camera-ready version.

**Limitations:**

The authors acknowledge the ellipsoidal approximation’s simplification of complex Rashomon sets but do not address critical limitations:
1. The method lacks analysis under non-convex or disjoint model geometries, which could break theoretical guarantees.
2. There is no discussion on whether counterfactuals might suggest unethical or harmful feature modifications. While the paper mentions "actionable" constraints, these are not explained.

**Paper Formatting Concerns:**

Line 73: lack of space.
Line 109: lack of equation number.
References to the Appendix are absent.

**Quality:**

3

**Strengths And Weaknesses:**

## Quality
### Strength
1. The work is technically rigorous, with well-formulated theorems. Authors provide closed-form solutions for worst-case models via ellipsoidal approximation. Authors present thorough empirical validation.
2. Claims are substantiated by experiments showing robustness improvements over baselines like T:Rex and Delta Robustness across linear models and neural networks.
3. The convex optimization formulation enables significant efficiency gains.
### Weaknesses:
1. The ellipsoidal approximation of the Rashomon set is a simplification that may fail in real-world scenarios where the true set has non-convex or disjoint geometry like multiple local optima. The authors acknowledge this but do not quantify error margins or evaluate performance under such conditions.
2. Experiments focus on linear models and neural networks with linearized final layer, leaving scalability to deep architectures untested.

---
## Clarity
### Strength
1. The paper is well-structured, with clear sections for background, methodology, theory, and evaluation. Mathematical formulation of ellipsoid derivation is precise.
### Weaknesses:
1. The Background and Notation section would benefit from better structure by splitting it into subsections and adding equation numbers, as it currently explains mathematical foundations in mostly plain text.
2. The transition from theoretical analysis to empirical validation in Section 6 is abrupt. A clearer roadmap connecting theorems to experimental results would enhance readability.
3. Technical intuition behind the Hessian-based approximation of the Rashomon set and how it captures model uncertainty is underdeveloped, leaving non-experts struggling to grasp the geometric reasoning.

---
## Significance
### Strength
1. The problem of robust counterfactual generation under model multiplicity is critical for high-stakes domains.
### Weaknesses:
1. While the paper emphasizes technical robustness, it lacks discussion on real-world deployment challenges, such as ensuring that suggested feature changes (e.g., increasing income) are actionable for users.
2. The societal implications of counterfactuals, particularly in cases where they might reinforce biases or suggest unethical modifications, are not addressed.

---
## Originality
### Strength
1. The method novel contribution lies in integrating convex geometry and robust optimization to generate theoretically guaranteed explanations under model uncertainty, a departure from ensemble-based or MILP methods.
2. The closed-form solution for worst-case models (Theorem 1) is particularly innovative.
### Weaknesses:
1. While the framework is original in its formulation, some ideas like parameter perturbation robustness [1], are inspired by prior work.
2. Authors could better explain how the method advances over existing methods that rely on ensembles or MILP [2], rather than geometry-based approximations.

[1] Jiang, Junqi, et al. "Formalising the robustness of counterfactual explanations for neural networks." Proceedings of the AAAI conference on artificial intelligence. Vol. 37. No. 12. 2023.

[2] Leofante, Francesco, Elena Botoeva, and Vineet Rajani. "Counterfactual explanations and model multiplicity: a relational verification view." Proceedings of the International Conference on Principles of Knowledge Representation and Reasoning. Vol. 19. No. 1. 2023.

---

> ### Author Rebuttal · Authors · 2025-07-31
>
> Thank you very much for the review and feedback.
>
> > Quality W1. The ellipsoidal approximation of the Rashomon set is a simplification that may fail in real-world scenarios
>
> The ellipsoidal approximation is a simplification of the Rashomon set that can be hard to find and fully approximate for a lot of datasets. However, our experiments are not limited to convex geometry only and effectively demonstrate that the approximation works and delivers robust results. More specifically, in Figure 1b, Table 1 and in Appendix C Figure 6 (Figure 11 for non data-supported) we use neural networks with cross-entropy loss and do not expect the loss to be convex. Therefore, we provide empirical evaluation in non-convex scenarios.
>
> >Q2 and  Quality W2. Experiments focus on linear models and neural networks with linearized final layer, leaving scalability to deep architectures untested.
>
> Our methodology is designed to apply to deep architectures by focusing its theoretical guarantees on the network's final linear layer. This layer operates on the embedding created by the preceding deep layers. This allows us to keep the robustness calculation tractable and fast.
>
> Once a worst-case model is computed in the embedding space using Theorem 1, the framework uses an optimization technique, such as the cutting-plane or gradient descent (we support both), to find the corresponding counterfactual in the original input space. This two-stage process allows the method to scale to deep networks, as our experiments confirm.
> We will explain this in more detail in the paper.
>
> > Clarity W1: The Background and Notation section would benefit from better structure.
>
> The background and notation section is only one page long, however, we will apply a modular structure and emphasize changes in the definitions better. We will add equation numbers to improve the paper's clarity.
>
> > Clarity W2: The transition from theoretical analysis to empirical validation in Section 6 is abrupt.
>
> Thank you, we will add better smoothing text when transitioning from Section 5 to 6.
>
> > Clarity W3: Technical intuition behind the Hessian-based approximation of the Rashomon set and how it captures model uncertainty is underdeveloped, leaving non-experts struggling to grasp the geometric reasoning.
>
> The core idea is to view the set of good models (the Rashomon set) as a "valley" in the loss landscape. Our method uses the Hessian matrix to mathematically describe the curvature or shape of this valley at its lowest point. A flat valley signifies high model uncertainty (many different models are good), while a steep valley indicates low uncertainty. The ellipsoid is a direct, geometric map of this curvature, providing a tractable representation of the model uncertainty that we use to find a provably robust counterfactual. We will add this more detailed explanation to the paper.
>
> Please note that Hessian-based approximations are a common technique used in various settings (see [1]-[3]).
>
> [1] Dinh, Laurent, et al. "Sharp minima can generalize for deep nets." International Conference on Machine Learning. PMLR, 2017.
>
> [2] Zhong, Chudi, et al. "Exploring and interacting with the set of good sparse generalized additive models." Advances in neural information processing systems 36 (2023): 56673-56699.
>
> [3] Sagun, Levent, et al. "Empirical analysis of the hessian of over-parametrized neural networks." arXiv preprint arXiv:1706.04454 (2017).
>
> > Limitation2 and Significance W1: While the paper emphasizes technical robustness, it lacks discussion on real-world deployment challenges, such as ensuring that suggested feature changes are actionable for users.
>
> Thank you for mentioning this. For the specific case when the users can not change some feature or they are immutable, such as for example, date of birth, we allow the users to select a set of features that they do not want to change (please see Appendix C.5.1 for more details). Technically, to obtain such counterfactuals, we block the gradient when optimizing with gradient descent for non-data-supported counterfactuals and search for counterfactuals that do not change immutable features for the data-supported version.
>
> Because we are focused on providing a robust and actionable recourse, we also **introduced a series of changes in order to make counterfactuals more actionable**.
>
> More specifically, we now additionally allow users to
>
> 1) Change of the feature in one direction (as if the value can only increase but not decrease, say if a person plans to increase income or spending amount when applying for a credit).
>
> 2) Fixed range for the feature.
>
> 3) Model for sparsity of the counterfactual explanations. We additionally employ L0 and L1 regularizations to allow for more sparse explanations for the cases when the user wants to change a small number of features.
>
> We believe that these changes should significantly improve the actionability of our explanations as well as user-experience while working with ElliCE.
>
> > Limitation2 and Significance W2: The societal implications of counterfactuals, particularly in cases where they might reinforce biases or suggest unethical modifications, are not addressed.
>
> While our work focuses on the technical challenge of providing reliable recourse, the resulting counterfactuals should be carefully audited in deployment to ensure they do not reinforce systemic biases. We will add this to our paper discussion and conclusions.
>
> > Originality W1: While the framework is original in its formulation, some ideas like parameter perturbation robustness [1], are inspired by prior work.
>
> Thank you for your comment. We carefully studied $Delta$-robustness in our paper both empirically and theoretically. Note that $Delta$-robustness works for neural networks only and introduces the perturbation to the whole neural network, while ElliCE uses ellipsoidal approximation in the embedding space for neural networks. In Appendix D (Proposition 1 and Theorem 6) , we theoretically show the limitations of the approach that $Delta$-robustness takes and show that it has limited ability to capture meaningful model multiplicity. Empirically, Ellice achieves better robustness guarantees, please see Figures 4, 6, 10, 11 in Appendix C.
>
> >  Originality W2: How the method advances over existing methods that rely on ensembles or MILP [2], rather than geometry-based approximations.
>
> We do not rely on ensembles or MIP. Instead, we propose a new method that relies on ellipsoidal approximation for the model multiplicity, gradient optimization for non-data supported counterfactuals and KD-tree and Ball-tree search for data-supported counterfactuals. Our method is much faster (Table 2) and more robust (Figures 4, 6, 10, 11 and Table 4 in Appendix C).
>
> > Limitation1 and Q1. How does your method perform when the true Rashomon set has non-convex or disjoint regions like multiple local optima?
>
> While the ellipsoidal approximation simplifies complex Rashomon sets, our empirical results show it remains effective even for non-convex cases. Our experiments with neural networks using cross-entropy loss (Figure 1b, Table 1, Appendix C Figures 6 and 11) demonstrate that ElliCE delivers robust counterfactuals despite non-convex loss landscapes.
>
> In cases with disjoint regions of good models, the ellipsoidal approximation doesn't fail but provides a local approximation around one optimum . In the worst case, this locality can lead to more conservative counterfactuals (due to the "price of robustness") or overly optimistic ones (which are not robust to the true, larger Rashomon set) . To account for multiple local minima, one could consider an ensemble of models trained with different random initializations. Note that we compared ElliCE against such a "Random Retrain" evaluation baseline and found ElliCE’s performance to be consistently strong, demonstrating its practical effectiveness even in the mentioned case.
>
>
>
>
> > Q3: Section 4 mentions "feature-level constraints". How are these constraints integrated into the optimization?
>
> In our framework, we allow for mutable and immutable features to handle feature-level constraints. More specifically, the user can tell the algorithms to fix some features and not change their values. We described this in Appendix C7. Implementation-based, we fix gradient flow for the gradient-based method and select counterfactuals that do not change the feature for tree-search optimization.
>
> After submission, we further improved our algorithms, and now we also handle:
>
> 1) change of the feature in one direction. This is implemented by blocking gradient components that would move features in the prohibited direction.  For the data-supported method, we do prefiltering.
>
> 2) fixed range for the feature. This is achieved by restricting gradient flow to stay within bounds and pruning KD-tree searches to exclude out-of-range solutions.
>
> 3) sparsity of the counterfactual explanations.  For data-supported counterfactuals, this is implemented by defining a custom distance metric for Ball tree search that combines sparsity and proximity ($metric = C \times \ell_0 + \ell_1$, where a coefficient C prioritizes minimizing the number of changed features).
>
> > Q4: What happens when a positive definite Hessian assumption fails (e.g. for for non-convex losses or high-dimensional data).
>
> The positive definite Hessian assumption is satisfied by design through regularization. If the Hessian is not positive definite, for instance, due to non-convexity or numerical issues, we perform additional regularization by adding a small multiple of the identity matrix, $\alpha*I$, to the computed Hessian. This standard technique guarantees the matrix becomes positive definite, ensuring numerical stability. This approach is effective regardless of data dimensionality or the non-convexity of the loss. We will make these details more explicit in the manuscript.
>
> > Formatting issues.
>
> Thank you for pointing out the formatting issues. We will fix them.

---

> > ### Comment · Reviewer_qf5X · 2025-08-05
> >
> > Thank you for your thorough rebuttal. I believe the presented work provides valuable theoretical and experimental contributions. I agree with reviewer cC5H that a submission to NeurIPS should minimize notation errors and adhere to proper academic writing standards. You have adequately addressed my questions and concerns. Assuming the authors will significantly improve the writing and clarity based on the provided feedback, I would like to increase my score to 5.

---

> > > ### Author Response · Authors · 2025-08-05
> > >
> > > Thank you for your feedback and for recognizing the contributions of our work. We will address comments to ensure a polished final version. We also sincerely appreciate your willingness to raise your score.

---

### Note · Authors · 2025-08-12

We thank all reviewers for their constructive feedback and thoughtful discussion. ElliCE introduces a theoretically grounded and computationally efficient framework for robust counterfactual explanations under model multiplicity, showing consistent improvements over baselines.

Post-rebuttal updates will focus on polishing presentation (clarity, notation, formatting), adding already-completed plausibility results and retraining/data-shift robustness results, expanding the technical intuition for the Hessian-based approximation and other assumptions made in the manuscript, clarifying societal implications and scalability to deeper architectures, and incorporating our recent improvements for more actionable recourse (e.g., immutable features, one-directional changes, bounded ranges, and sparsity controls).

All requested clarifications have been addressed during discussion, and we will incorporate them into the final version to ensure the paper is clearly communicated. Thank you for helping us improve our work.

---

### Decision · Program_Chairs · 2025-09-17

**Decision:**

Accept (spotlight)

**Comment:**

The paper proposes ElliCE, a method for generating counterfactual explanations robust to model multiplicity by approximating the Rashomon set with an ellipsoid, offering provable guarantees and strong empirical results.  Its main strengths are the novel geometric formulation, clear theoretical contributions, and efficiency gains.  Weaknesses include reliance on simplifying assumptions, limited theoretical scope beyond linearized models, and incomplete treatment of plausibility and societal implications, alongside clarity issues.  The decision to accept rests on the significance of providing provably robust and efficient recourse, which outweighs these shortcomings.  During discussion, concerns about non-convexity, scalability, plausibility, and clarity were largely resolved by the rebuttal, with reviewers raising their scores.